# Emergence of a smooth interface from growth of a dendritic network against a mechanosensitive contractile material

**Medha Sharma, Tao Jiang, Zi Chen Jiang, Carlos E Moguel-Lehmer, Tony J C Harris***

Department of Cell and Systems Biology, University of Toronto, Toronto, Canada

**Abstract** Structures and machines require smoothening of raw materials. Self-organized smoothening guides cell and tissue morphogenesis and is relevant to advanced manufacturing. Across the syncytial *Drosophila* embryo surface, smooth interfaces form between expanding Arp2/3-based actin caps and surrounding actomyosin networks, demarcating the circumferences of nascent dome-like compartments used for pseudocleavage. We found that forming a smooth and circular boundary of the surrounding actomyosin domain requires Arp2/3 in vivo. To dissect the physical basis of this requirement, we reconstituted the interacting networks using node-based models. In simulations of actomyosin networks with local clearances in place of Arp2/3 domains, rough boundaries persisted when myosin contractility was low. With addition of expanding Arp2/3 network domains, myosin domain boundaries failed to smoothen, but accumulated myosin nodes and tension. After incorporating actomyosin mechanosensitivity, Arp2/3 network growth locally induced a surrounding contractile actomyosin ring that smoothened the interface between the cytoskeletal domains, an effect also evident in vivo. In this way, a smooth structure can emerge from the lateral interaction of irregular active materials.

**\*For correspondence:**
tony.harris@utoronto.ca

**Competing interests:** The authors declare that no competing interests exist.

## Introduction

Smooth components are integral to the assembly and function of human-made structures and devices. Smooth surfaces dominate our inside and urban environments and are formed by refinement of rough starting materials. In conventional industries, starting materials are smoothened by tools. Smooth structures also convey important functions in living systems, but their formation relies on self-assembly and self-organization. Fuller understanding of these mechanisms will aid advanced manufacturing with smart materials for a broad range of applications; from sensors and data processing, to filters and self-cleaning surfaces, to medical devices and robotics (*Begley et al., 2019*; *Eder et al., 2018*; *Holmes, 2019*; *Wang et al., 2020*).

Living and synthetic systems face a challenge: how can small and roughly distributed components form large and smooth structures that convey function? One solution is stereospecific self-assembly of components into a structure with mechanical properties that convey smoothness. For example, tubulin subunits assemble microtubules with relatively high rigidity, allowing microtubules to act as smooth tracks for molecular transport through the cell (*Hawkins et al., 2010*). Individual actin polymers have lower rigidity (*Salbreux et al., 2012*), but cross-linking by actin-binding proteins can bundle multiple polymers into straight structures that support cell protrusions (*Svitkina, 2018*). In more complex scenarios, solutions involve an initial step of rough patterning followed by refinement through mechanical self-organization. One example is the establishment of a compartment boundary across an epithelial tissue. In the *Drosophila* wing imaginal disc, gene expression patterns define the cells of specific compartments, but non-muscle myosin II (myosin hereafter) is then required to pull actin networks into straight cables integrated via cell-cell junctions to form smooth compartment

boundaries that define the architecture of the mature wing (*Harris, 2018*; *Wang and Dahmann, 2020*). Another example comes from mesenchymal cell migration. Here, a leader cell's initial path through the extracellular matrix is smoothened by follower cells as reciprocal mechanical effects co-align cellular actomyosin cables with extracellular matrix fibers via focal adhesions (*Livne and Geiger, 2016*; *van Helvert et al., 2018*). A purely subcellular example occurs during cytokinesis, when spindle-derived cues roughly organize actomyosin assemblies around the cell equator, and myosin activity then contributes to smoothening of the contractile cytokinetic ring that divides the cell in two (*Schwayer et al., 2016*). 'Two-step' refinement strategies are also relevant to hierarchical design principles important for transcending scales in advanced manufacturing (*Begley et al., 2019*; *Eder et al., 2018*; *Holmes, 2019*; *Wang et al., 2020*).

In addition to specialized structures, such as junction-associated cables or cytokinetic rings, actin networks also form the cell cortex, a thin, sheet-like material underlying the cell's plasma membrane (*Chugh and Paluch, 2018*; *Svitkina, 2020*). Smooth structures can form across the cell cortex, but the mechanisms involved are not fully understood. Two properties of the cortex are relevant: (i) it can gain an asymmetric distribution across the cell and (ii) it is a composite material. Cortical asymmetry can elicit cortical flow (*Munro, 2006*), with resulting compressive forces promoting mesoscale alignment of actin filaments perpendicular to the flow (*Reymann et al., 2016*). The cortex's composite structure involves distinct actin-based networks: formin-induced networks, with or without myosin, and Arp2/3-induced networks (*Chugh and Paluch, 2018*; *Svitkina, 2020*). These networks can be closely interwoven, but signaling events can induce mesoscale cortical domains enriched with a particular network. For example, the budding of a polar body from a much larger oocyte involves induction of an Arp2/3-enriched domain surrounded by a smooth ring of actomyosin (*Yi and Li, 2012*). The Arp2/3-enriched domain prevents full contraction of the actomyosin ring, and the actomyosin ring delineates the boundary of budding polar body, but how a smooth interface forms between such networks remains unclear.

The cortex of the early *Drosophila* embryo is patterned into subdomains that engage each other to organize embryo pseudocleavage. Centrosomes provide local spatial cues that induce an Arp2/3-enriched actin cap above each nucleus (*Foe et al., 2000*; *Raff and Glover, 1989*; *Stevenson et al., 2002*; *Stevenson et al., 2001*; *Zallen et al., 2002*), and each cap is surrounded by an actomyosin border (*Foe et al., 2000*). Since the early embryo is a syncytium, the cortical actomyosin network is embryo-wide and becomes embedded with numerous, evenly spaced caps. At the onset of each synchronous mitotic cycle, the caps and borders assemble anew, and expand as cortical domains with rough boundaries (*Foe et al., 2000*; *Zhang et al., 2018*). As the caps and borders meet, they form a smooth, circular interface that demarcates the circumference of a nascent dome-like compartment. Each cap grows into a full dome-like compartment that houses the mitotic spindle, and the cap-border interface becomes the basal rim of the dome. The networks grow, meet, form a circular interface, and locally re-shape the cortex within minutes. Associated cortical patterning involves a two-step refinement process. Initial assembly of each domain is induced by signaling of distinct small G proteins (*Blake-Hedges and Megraw, 2019*; *Lv et al., 2021*). Subsequently, the interface between the domains seems to be refined through physical mechanisms. For example, with experimental depletion of one network, the other network expands its domain laterally due to an apparent loss of physical restriction (*Zhang et al., 2018*). Each network also contributes to the structure of the dome-like compartment. The Arp2/3 network is necessary for compartment formation (*Stevenson et al., 2002*; *Zallen et al., 2002*), and when myosin levels or activity are experimentally reduced, growth and buckling of the actin cap appears sufficient to form a compartment (*Royou et al., 2004*; *Zhang et al., 2018*). With myosin depletion, however, the actin cap grows with an irregular shape, and the dome-like compartment is abnormally lop-sided (*Zhang et al., 2018*). Thus, lateral engagement of a growing Arp2/3 cap with its surrounding actomyosin border seems required for precise re-shaping of the cell cortex into a dome-like compartment, but physical explanations remain unclear. We sought to understand the interface between these materials. Specifically, we pursued how the initially rough-edged networks interact to increase the smoothness and circularity of their interface. We considered smoothening in relative terms: an increase to the alignment of cytoskeletal elements at the boundary of a network. Functionally, this alignment forms a mesoscale circular structure where the cell surface folds during *Drosophila* pseudocleavage.

Here, we provide in vivo evidence that Arp2/3 is required for forming a smooth, circular interface between an actin cap and its actomyosin border, and for myosin accumulation at the network-

network interface. To study the networks in isolation, we turned to node-based simulations. Individually reconstituted actomyosin and Arp2/3 networks mimicked mesoscale properties reported for each network in vitro. The node-based design of the networks allowed them to be combined in the same simulations, in which they were distributed as segregated domains and were capable of displacing each other laterally. Without simulated Arp2/3 networks, high myosin activity produced smooth, circular boundaries with empty caps, whereas low myosin activity resulted in persistent rough boundaries, mimicking the in vivo effect of *Arp3* RNAi. Added Arp2/3 caps displaced the myosin borders, but the resulting interfaces were wavy and non-circular, despite local increases to myosin node density and tension. By encoding a mechanosensitive activation mechanism into the actomyosin network, growth of an Arp2/3 cap induced myosin activations around the interface, an actomyosin ring formed, and a smooth and circular interface emerged. Our findings show how an interface can smoothen from the interaction of roughly distributed active materials in a 2D plane.

## Results

Actomyosin and Arp2/3 networks have been individually simulated as continuum or agent-based models (*Berro, 2018*; *Cortes et al., 2018*; *Mogilner and Manhart, 2016*). Our investigation of roughly distributed elements rearranging into a smooth structure required an agent-based approach. We formulated 2D node-based models in which actomyosin network nodes displace each other with the contractile force of myosin, and Arp2/3 network nodes are displaced by the polymerization force of actin (*Banerjee et al., 2020*; *Lecuit et al., 2011*). The initial actomyosin network pattern was laid out by distributing myosin nodes across a 2D plane. In local search areas, each node then randomly established a limited number of permanent connections with surrounding nodes. To mimic the thickness of the cell cortex, connections were allowed to cross each other. Myosin nodes affected each other by applying contractile force along their connections, and connections experienced elastic force above their resting length (see *Figure 1A* for schematics; see Model Formulation for full details and references). Arp2/3 networks were seeded with clusters of actin nucleation points, and a plus end node moved from each of these sites in random directions within the 2D plane. At a specific time, a new actin nucleation point formed halfway along the plus end node's trajectory, and a new plus end node grew from this site at a 70° angle. Plus end nodes stopped their movement at a specified distance from their coupled actin nucleation points, and were eliminated at later times with a specific probability. Actin nucleation points were only used to initiate plus end node movements, and to determine their trajectories. Plus end nodes affected myosin nodes with polymerization force (see *Figure 1F* for schematics; see Model Formulation for full details and references). A common node-based construction of the networks facilitated modelling of their lateral interface. Segregation of the networks was maintained at the network-network interface, where the forces of one network's nodes could displace the nodes of the other based on local force balance (see Model Formulation for full details). Our model is based on relatively simple physical rules applied to individual nodes, and we used it to investigate the emergence of mesoscale structure by simulating, visualizing, and quantifying large networks of nodes. *Table 1* provides a full list of model parameters and estimates of in vivo and in vitro equivalencies. Before investigating network-network interactions, we first validated the in silico properties of the actomyosin network alone, the Arp2/3 network alone, and the network-network interface.

### The actomyosin network model mimics properties of actomyosin networks reported in vitro

Since we sought to understand the emergence of a mesoscale property, a smooth interface between two cortical networks, we validated the simulated actomyosin network by comparing its mesoscale properties with those of an actomyosin network reconstituted in vitro. By inducing myosin motor activity in specific geometric patterns within a larger non-contractile network in vitro, *Schuppler et al., 2016* showed that re-shaping of the embedded contractile actomyosin network is influenced by the boundary pattern of its initial activation (*Schuppler et al., 2016*). As a simple example, an activated circle of actomyosin contracted into a smaller circle. More interestingly, the edges of an activated square became concave, whereas the edges of an activated outline of a square became convex (*Schuppler et al., 2016*). Remarkably, optogenetic activation of actomyosin in the complex environment of the *Drosophila* embryo ectoderm had a similar effect: the edges of

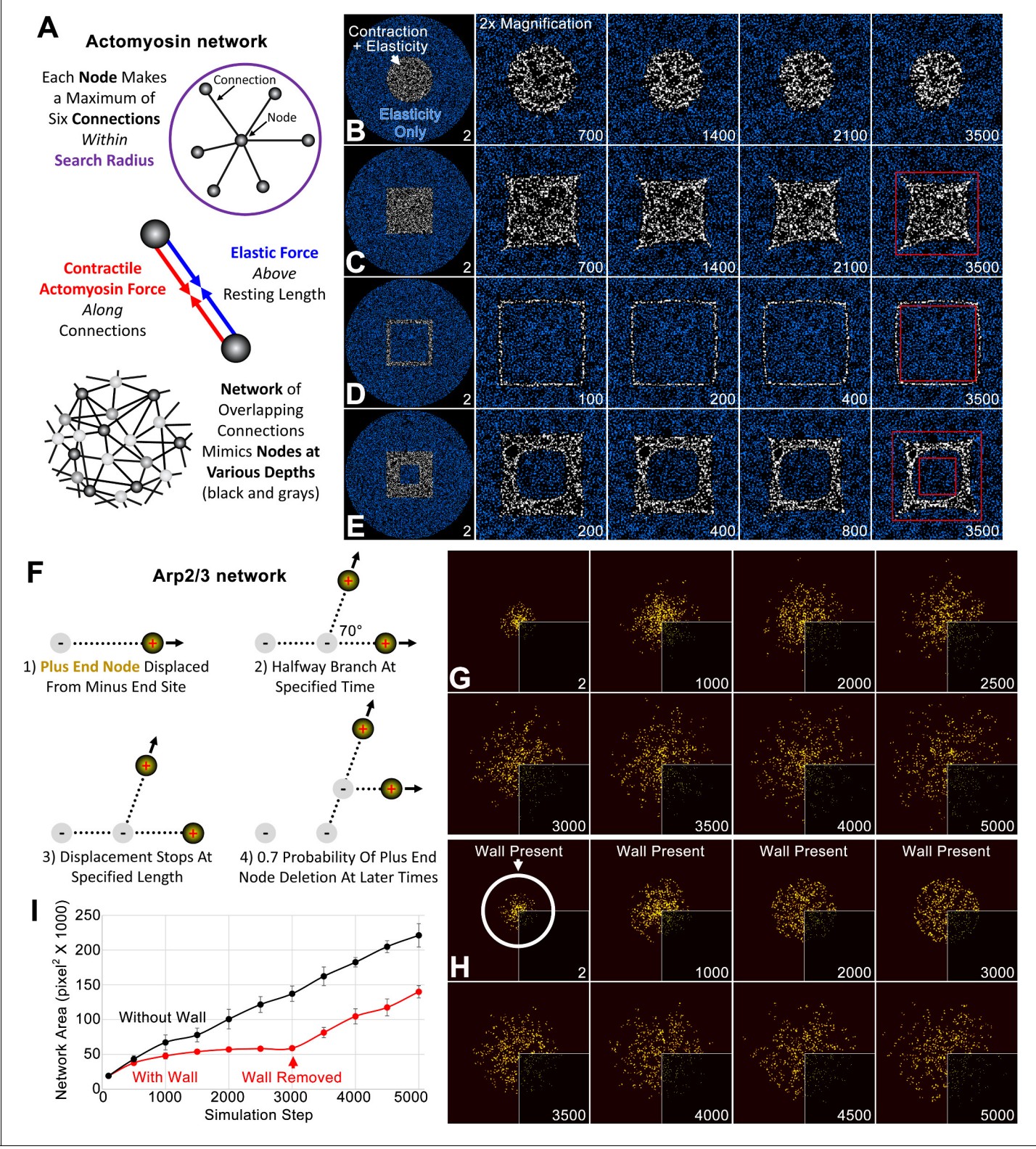

**Figure 1.** Individual simulations of actomyosin and Arp2/3 networks. (**A**) Schematics show major aspects of the node-based actomyosin network: (top) the connections randomly made by one node within a search radius, (middle) the actomyosin and elastic forces along node-node connections, (bottom) the organization of nodes and connections mimicking a broad cortical network with a small thickness. (**B–E**) Within a large circle of nodes with only elastic connections (blue), node-based contractility is additionally induced in specific patterns (white). Twofold magnifications of the central region show

*Figure 1 continued on next page*

*Figure 1 continued*

the shape changes of the activated regions over indicated simulation steps (bottom right of each panel). Red squares provide references to detect the shape changes in C-E. In each case, similar results were observed for five simulations with distinct starting conditions. (F) Schematics show major aspects of the node-based Arp2/3 network, focusing on (1) the displacement of one plus end node from a nucleation point (minus end), (2) branching from this connection, (3) halting of the plus end node displacement, and (4) deletion of the plus end node. (G) Growth of plus end nodes (yellow) from a central patch of nucleation points over indicated simulation steps. Bottom right quadrants show the original simulation images and visualization of the other quadrants was enhanced by Gaussian blurring and a brightness and contrast adjustment. (H) The setup and visualization is the same as (G) except a circular wall was in place until simulation step 3000 (the position of the wall is shown at simulation step 2). (I) Network area plotted over simulation steps without a wall (black) and with a wall from the start of the simulation to simulation step 3000 (red). Note the similar area growth rate of both cases after simulation step 3000. Means ± SD shown for four simulations with distinct starting conditions.

an activated square became concave as the square contracted (*Izquierdo et al., 2018*). To test our simulated actomyosin network, we activated myosin node contractility in different shapes embedded within a thick border of nodes that lacked myosin node contractility but retained elastic connections (*Figure 1B–E*; see Model Formulation for details). Not surprisingly, an activated circle displayed isotropic contraction (*Figure 1B*). Notably, the sides of an activated solid square became concave (*Figure 1C*), mimicking both the in vitro (*Schuppler et al., 2016*) and in vivo studies (*Izquierdo et al., 2018*). The outer sides of a hollow square with a thin boundary of activation became convex (*Figure 1D*), mimicking the in vitro behavior (*Schuppler et al., 2016*). For a hollow square with a thick boundary of activation, the outer sides of the boundary became concave and the inner edges bent in the opposite direction (*Figure 1E*), as also seen in vitro (*Schuppler et al., 2016*). In addition to the patterned activation of myosin, these realistic responses also required the entire simulated network to contain elastic connections that prevented excessive stretching of connections between the inner contractile region and the outer non-contractile region.

## The Arp2/3 network model mimics properties of Arp2/3 networks reported in vitro

When simulated individually, a centrally induced Arp2/3 network grows centrifugally with an outer perimeter of non-aligned actin plus end nodes (*Figure 1G*), a behavior resembling initial in vivo growth of an actin cap in the embryo surface plane (e.g. *Zhang et al., 2018*). Next, we compared the simulated network to an Arp2/3 actin network reconstituted in vitro and grown against a solid object. By combining light and atomic force microscopy, *Bieling et al., 2016* found that Arp2/3 network growth against a cantilever lead to an increased density of F-actin plus ends at the interface, without an accumulation of elastic energy (*Bieling et al., 2016*). We tested our model for these properties by placing a circular wall around a centrifugally growing network. With network growth into the immovable wall, the density of plus end nodes within 20 pixels of the network circumference increased 1.57-fold from the start of the simulation to step 3000 (calculated from four simulations with distinct starting conditions; see images in *Figure 1H*). For networks growing without a wall, the equivalent value decreased 0.77-fold by simulation step 3000 (see example images in *Figure 1G*). To test for stored elastic energy, we removed the wall after local accumulation of plus end nodes had occurred. If elastic energy accumulated with compression, then an increased network growth rate upon removal of the load was expected. However, network area growth rates after removal of the wall were indistinguishable from those of networks that never experienced a wall (*Figure 1I*). Thus, like an in vitro Arp2/3 network (*Bieling et al., 2016*), when the simulated network grew against a solid object it accumulated plus end numbers at the interface but did not accumulate elastic energy.

## Simulation of the network-network interface maintains network segregation and allows the networks to displace each other

To test the mesoscale effects of our internetwork displacement and segregation rules, we distributed myosin nodes as a broad ring and examined behavior of the ring with and without a growing Arp2/3 network at its hollow center. In the absence of a central Arp2/3 network, the ring was reshaped by myosin activity alone. Without myosin activity (0.0), the broad ring maintained its structure (*Figure 2A*). With full myosin activity (1.0), the broad ring of dispersed nodes initially constricted into a narrow ring of nodes, and after thinning, the ring constricted centripetally (*Figure 2E*), a

**Table 1.** Parameters used in simulations.

| Parameter | Value | Remark |
|---|---|---|
| One simulation step | 0.1 s | Estimated from in vivo cap growth (*Jiang and Harris, 2019*) |
| One pixel | 0.1 micron | Estimated from in vivo cap growth (*Jiang and Harris, 2019*) |
| **Actomyosin network** | | |
| Myosin node density ($Den\_m$) | Varied (0–1) | Proportion of total pixels in the actomyosin zone with a myosin node |
| Myosin node activity (M) | Varied (0–1) | Relative activities |
| Search radius for establishing connections with surrounding myosin nodes ($D_{thres}$) | five pixels | Based on optimal actin filament length for cortical actomyosin contractility (*Chugh et al., 2017*) |
| Maximum number of connections with surrounding myosin nodes ($Max_{Neigh}$) | 6 | Optimized* |
| Maximum number of myosin nodes per pixel ($N_{max}$) | 20 | Optimized* |
| Coefficient of myosin force ($K_{myosin}$) | 0.5 nN | Order of magnitude measured for myosin force (*Finer et al., 1994*) |
| Coefficient of spring force ($K_{spring}$) | 0.05 nN/nm | Order of magnitude measured for actin elasticity (*Kojima et al., 1994*) |
| Node-node connection resting length ($l_o$) | Variable | Mean length of all connections before a simulation starts |
| **Arp2/3 network** | | |
| Initial nucleation site density ($Den\_c$) | 0.1 and 0.01[‡] | Proportion of total pixels in the nucleation zone with a nucleation site[†] |
| Actin polymerization force coefficient (Kpoly) | 0.5 nN | Order of magnitude measured for Arp2/3 networks with same area (*Bieling et al., 2016*)[†] |
| Age of plus end node when a branch is induced ($Age_{Br}$) | nine steps | Optimized* |
| Maximum length between nucleation site and plus end node (Lth) | 20 pixels | Ordered of magnitude calculated for capped actin filaments (*Schafer et al., 1996*) |
| Minimum age of plus end node when its loss becomes possible ($Age_{th}$) | 20 steps | Optimized* |
| Probability of loss of plus end node at each step (Pdel) | 0.7 | Optimized* |
| **Network-network Interface** | | |
| Radius around a node's initial position to identify potentially interacting nodes of the other network (SR) | five pixels | Assumed[‡] |
| Radius around a node's target position to determine if nodes of the other network are absent, which allows the move ($V_{ex}$) | two pixels | Assumed[‡] |

*Optimized in relation to other parameters of the actomyosin or Arp2/3 model (see Model Formulation for details).

[†]These parameters of the Arp2/3 model were coarsened to account for a plus end node representing only one actin filament whereas a myosin node represents a myosin mini-filament connected to many actin filaments. To increase the impact of each plus end node relative to a myosin node, the initial actin nucleation site density was set on the lower side whereas the polymerization force of a single plus end node was set on the higher side.

[‡]See Model Formulation for details.

sequence of events consistent with in vitro observations of a myosin network activated in a rectangular shape (*Schuppler et al., 2016*). A moderate level of myosin activity (0.4) had an intermediate effect (*Figure 2C*). Adding a growing Arp2/3 network to the center of a broad ring with zero or moderate myosin activity led to centrifugal displacement of myosin nodes and a thinning of the ring, but no ring constriction was observed (*Figure 2B,D*). As the Arp2/3 network grew, the thinned ring of myosin nodes was displaced outward and displayed diminished circularity that resembled the rough outer edge of the segregated Arp2/3 network. During this displacement, the two networks remained segregated, confirming the effectiveness of the node-based rules, and myosin nodes aggregated at the network-network interface (*Figure 2B,D*, arrows). With full myosin node activity, the actomyosin ring counteracted the expansion of the central Arp2/3 network (*Figure 2F*). The ring narrowed and then contracted centripetally, but the central Arp2/3 network restrained its centripetal

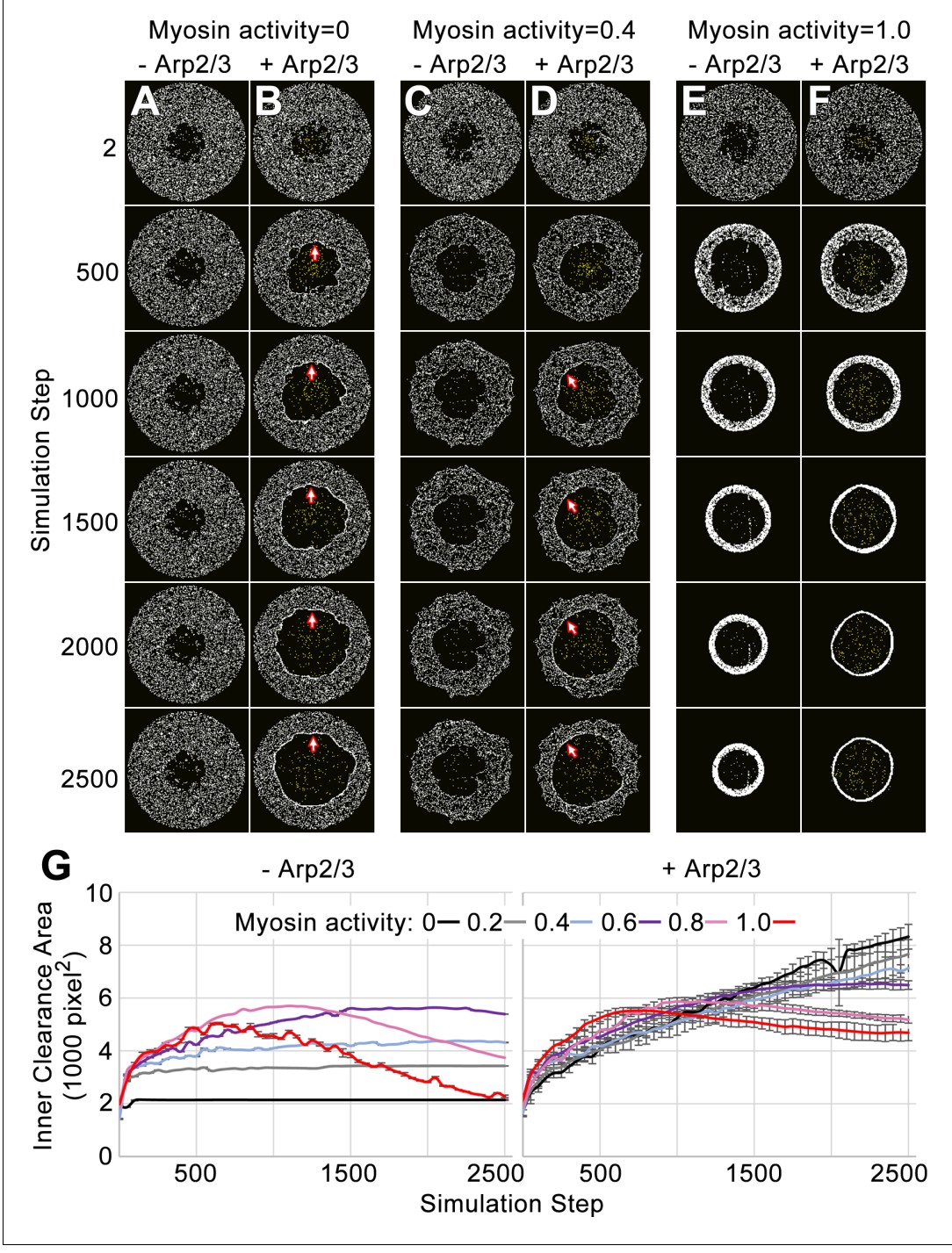

**Figure 2.** Simulation of the network-network interface allows the networks to displace each other and remain segregated. (A–F) Actomyosin networks (white) with zero (A–B), moderate (0.4) (C–D), or full (1.0) (E–F) myosin activity. In each case, the initial configuration is a circle with an inner clearance of nodes removed in a rough pattern. Each case is simulated with (B, D, F) or without (A, C, E) an Arp2/3 network grown from a central patch of nucleation points within the inner clearance (Arp2/3 networks shown in yellow). (A) With zero myosin activity, the actomyosin network maintains its shape without an Arp2/3 network. (B) In response to Arp2/3 network growth, the inner area expands and myosin nodes aggregate at the network-network interface (arrows). (E) With full myosin activity (1.0) in the absence of an Arp2/3 network, the thickness of the actomyosin ring first decreases (accompanied by an increase to the inner clearance area and an increased density of myosin nodes in the ring) and then the overall ring constricts (accompanied by a decrease to the inner circle area). (F) With full myosin

*Figure 2 continued on next page*

*Figure 2 continued*

activity (1.0) in the presence of a growing Arp2/3 network, the actomyosin ring undergoes greater thinning and then the overall constriction of the ring is reduced. (**C–D**) Simulations of moderately contractile actomyosin networks (0.4 activity) alone or with a growing Arp2/3 network, displayed intermediate behaviors. (**G**) Quantifications of inner clearance area versus simulation step for actomyosin networks with a range of activities (0–1.0), without Arp2/3 networks (left) or with Arp2/3 networks (right). Means ± SD shown for three simulations with distinct starting conditions. SD values were low without addition of the growing Arp2/3 networks.

contraction relative to an actomyosin ring lacking a central Arp2/3 network (*Figure 2F*). Quantifications of inner clearance area changes across myosin node activities with or with Arp2/3 networks (*Figure 2G*) further demonstrated that the rules for the network-network interface allowed the two segregated networks to counteract each other's physical effects. For example, without myosin activity, the inner clearance area was maintained over the simulation, but adding an Arp2/3 network to inner clearance expanded its area (*Figure 2G*, compare black lines with and without Arp2/3). With full myosin activity, the effect of the Arp2/3 network was counteracted (*Figure 2G*, compare black and red lines in right graph), but the Arp2/3 network also hindered the contractility of the myosin ring (*Figure 2G*, compare red lines with and without Arp2/3).

## A smooth and circular interface between an actin cap and its actomyosin border requires Arp3 in vivo

Genetic analyses of *Drosophila* embryo cleavage have shown that Arp2/3 is required for displacing actomyosin networks from the cap and that myosin activity is needed for restraining lateral cap growth. These and other data suggest that the two networks physically engage to control each other's distributions (*Zhang et al., 2018*). A physical interaction should be evident at the interface between the laterally segregated networks. To investigate the effect of Arp2/3 on myosin organization at this interface, we imaged GFP-tagged myosin heavy chain (Zipper (Zip)-GFP) in embryos depleted of Arp3 by RNAi. In control RNAi embryos during the transition from mitotic cycle 10 to 11, the actomyosin network initially assembled a rough, non-circular interface around each myosin-devoid cap, but within minutes, smooth and circular boundaries formed (*Figure 3A*; the transition was most obvious for completely new networks formed between sister nuclei, solid black arrows). In contrast, Arp3 RNAi embryos assembled actomyosin networks that failed to form smooth boundaries and instead retained a variable distribution of puncta at the interface with the myosin-devoid cap (*Figure 3A*, solid red arrows). In control embryos, the formation of a smooth interface was often accompanied by an accumulation of Zip-GFP along the interface (*Figure 3A*, solid and hollow black arrows), and both demarked the site of initial pseudo-cleavage furrow ingression (*Figure 3A*, 143s, side view, blue arrowheads). Arp3 RNAi embryos lacked Zip-GFP accumulation at the interface (*Figure 3A*, solid and hollow red arrows) and furrow ingression failed (*Figure 3A*, 143s, side view, blue bracket shows a broad band of myosin at the embryo surface). These data suggest that the Arp2/3-based actin network of the cap is required for forming a smooth interface with the actomyosin border, and that smooth interface formation coincides with myosin accumulation and furrow initiation at the interface. Notably, both networks initially have a rough boundary (*Zhang et al., 2018*; *Figure 3A*), and the smooth interface seems to arise from their lateral interaction.

## The effect of depleting Arp2/3 in vivo is mimicked by simulations with no Arp2/3 networks and low myosin activity

To address how a smooth boundary forms in vivo, we configured our simulated actomyosin and Arp2/3 networks to resemble their arrangement in the embryo. The embryo cortex is coated with a continuous actomyosin network embedded with evenly distributed Arp2/3-based actin caps (*Foe et al., 2000*; *Zhang et al., 2018*; *Figure 3A*). This organization was simulated by generating a large circle of active myosin nodes with 13 evenly spaced clearances for the growth of 13 Arp2/3 networks (*Figure 3B*). Within the large circle, the myosin nodes were activatable in addition to their elastic interactions. Outside of the large circle, a border of inactive but elastic myosin nodes was generated to restrain the contraction of the central circle, thereby mimicking the effect of an actomyosin network distributed around the full embryo circumference. To avoid distortions due to the

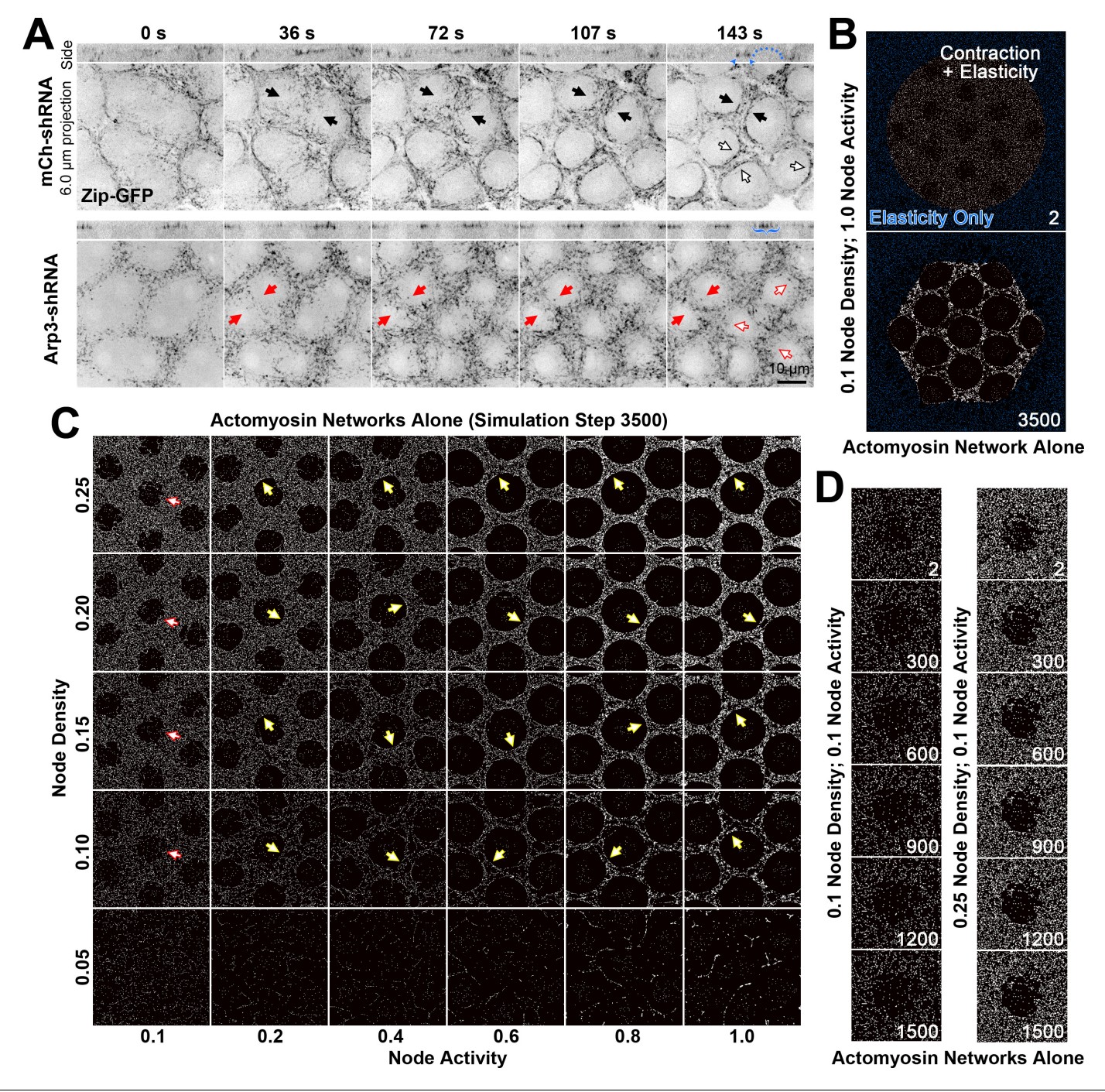

**Figure 3.** Actomyosin network organization in the absence of Arp2/3 networks in vivo and in silico. (**A**) In vivo live imaging of Zip-GFP (myosin) from telophase of cycle 10 to prophase of cycle 11 in control (mCh-shRNA) and Arp3 RNAi (Arp3-shRNA) embryos. In the control embryo, solid black arrows show the rough-to-smooth transitions of myosin network boundaries from 36 s to 143 s. By 143 s, the boundaries display an accumulation of Zip-GFP (solid and hollow black arrows), and doming of the cortex (blue dots) is evident in the side view with myosin accumulations at the base of the dome (blue arrowheads). In the Arp3 RNAi embryo, the boundaries of the myosin network remain rough (red arrows) and fail to accumulate Zip-GFP, which retains a punctate distribution (red solid and hollow arrows). In the side views, the cortex remains flat with broader distributions of Zip-GFP (blue brackets). These observations were made in 15/15 control embryos and 8/9 Arp3 RNAi embryos (additional examples shown in *Figure 3—figure supplement 1*). (**B**) An example of the configuration used to simulate the in vivo organization of the actomyosin network. At the start of the simulation, a large circle of myosin nodes with contractility and elasticity (white) is surrounded by a square boundary of myosin nodes with only elasticity (blue). Within the large contractile circle, 13 roughed-edged clearances were laid out at the beginning of the simulation. Simulation step 3500 shows the

*Figure 3 continued on next page*

*Figure 3 continued*

contracted state of the simulation with high myosin activity. (C) A phase diagram of increasing myosin node densities compared with increasing myosin node activities. The central seven clearances of configurations similar to (B) are shown at simulation step 3500. Note the network fragmentation at node densities of 0.05 (bottom row). Yellow arrows indicate smooth boundary formation at node densities between 0.10 and 0.25 and myosin activities between 0.2 and 1.0. Red arrows indicate rough boundary maintenance at node densities between 0.10 and 0.25 and myosin activity of 0.1. (D) At 0.1 myosin activity, network shapes change initially (step 2–300) but are then maintained. The central clearance of configurations similar to (B) are shown for node densities of 0.1 and 0.25. The observations of (C) and (D) were reproduced in three sets of simulations with distinct starting conditions.

The online version of this article includes the following figure supplement(s) for figure 3:

**Figure supplement 1.** Multiple examples of myosin organization in control and Arp3 RNAi embryos at cycle 11 prophase.

surrounding inactive border, we restricted our quantifications to the central clearance of the 13-clearance array.

To understand the effects of actomyosin alone, we first varied properties of the myosin nodes in the absence of Arp2/3 networks. The density of nodes was varied over a fivefold range, and the activity of nodes was varied over a 10-fold range (*Figure 3C*). Within each simulation, all nodes had homogenous activity. The lowest density of nodes (0.05; 5% of pixels containing a node) resulted in fragmentation of the network at all myosin activity levels (*Figure 3C*; bottom row), consistent with in vitro studies (*Schuppler et al., 2016*). At node densities that retained connectivity (0.1–0.25), higher myosin activities (0.2–1.0) produced smooth and circular boundaries with the clearances (*Figure 3C*, yellow arrows). This result is consistent with the sufficiency of myosin activity to produce extended, smooth structures in vivo, such as cell junction-associated actomyosin cables (*Harris, 2018*; *Livne and Geiger, 2016*) or the cytokinetic ring (*Schwayer et al., 2016*). In contrast, low myosin activity (0.1) allowed persistence of rough boundaries with the clearances (*Figure 3C*; red arrows), and displayed modest network rearrangements during early stages of the simulation (*Figure 3D*). This result resembled the myosin distribution in Arp3 RNAi embryos (*Figure 3A*). Together, these data suggest that myosin activity is relatively low in the early embryo. Indeed, several inhibitors of myosin activity are required for maintaining proper syncytial architecture of the early *Drosophila* embryo (*Lee and Harris, 2013*; *Mason et al., 2016*; *Zhang et al., 2018*), and a specific mechanism locally elevates myosin activity for budding and full division of mono-nucleated primordial germ cells from the syncytium (*Cinalli and Lehmann, 2013*). The simulated myosin networks with enough density for broad connectivity but insufficient activity to form smooth boundaries provided an opportunity to study the effect of adding Arp2/3-based actin caps.

## Arp2/3 network growth is insufficient to form a smooth and circular interface with a weakly contractile myosin network

To test the effect of centrifugal cap growth, we induced Arp2/3 network growth from the center of each clearance within weakly contractile actomyosin networks (0.1 activity) with two different myosin node densities (0.1 and 0.2). Compared with the myosin network boundaries formed in the absence of Arp2/3 networks (*Figure 3C–D*), local smoothness improved but the network-network interface was wavy, and each clearance gained a hexagon-like shape (*Figure 4A*). Sensitivity analyses showed a failure to form a smooth and circular interface over a wide range of parameter values affecting the myosin network, the Arp2/3 network, and their interface (*Figure 4—figure supplements 1–6*). However, Arp2/3 network growth did induce an aggregation of myosin nodes at the interface (*Figure 4A*, arrows). Also, myosin node connections became substantially longer than their resting length specifically at the interface, and only with Arp2/3 network growth (*Figure 4B*). In the presence of growing Arp2/3 networks, node-node connections of two or more pixels greater than resting length rose linearly from 0% of all connections at simulation step 0, to 0.48 ± 0.01% at step 500, to 2.05 ± 0.06% at step 2500 (calculated from four simulations with distinct starting distributions, a myosin node density of 0.2, and a myosin node activity of 0.1; p<0.001 comparing steps 500 and 2500). In the absence of growing caps, the percentages remained at <0.003% from simulation steps 0 to 2500. Thus, Arp2/3 network growth aggregated myosin nodes at the interface and increased tension between myosin nodes at the interface. However, it also resulted in a wavy interface and a space-filling effect, suggesting the simulated actomyosin borders were unnaturally deformable.

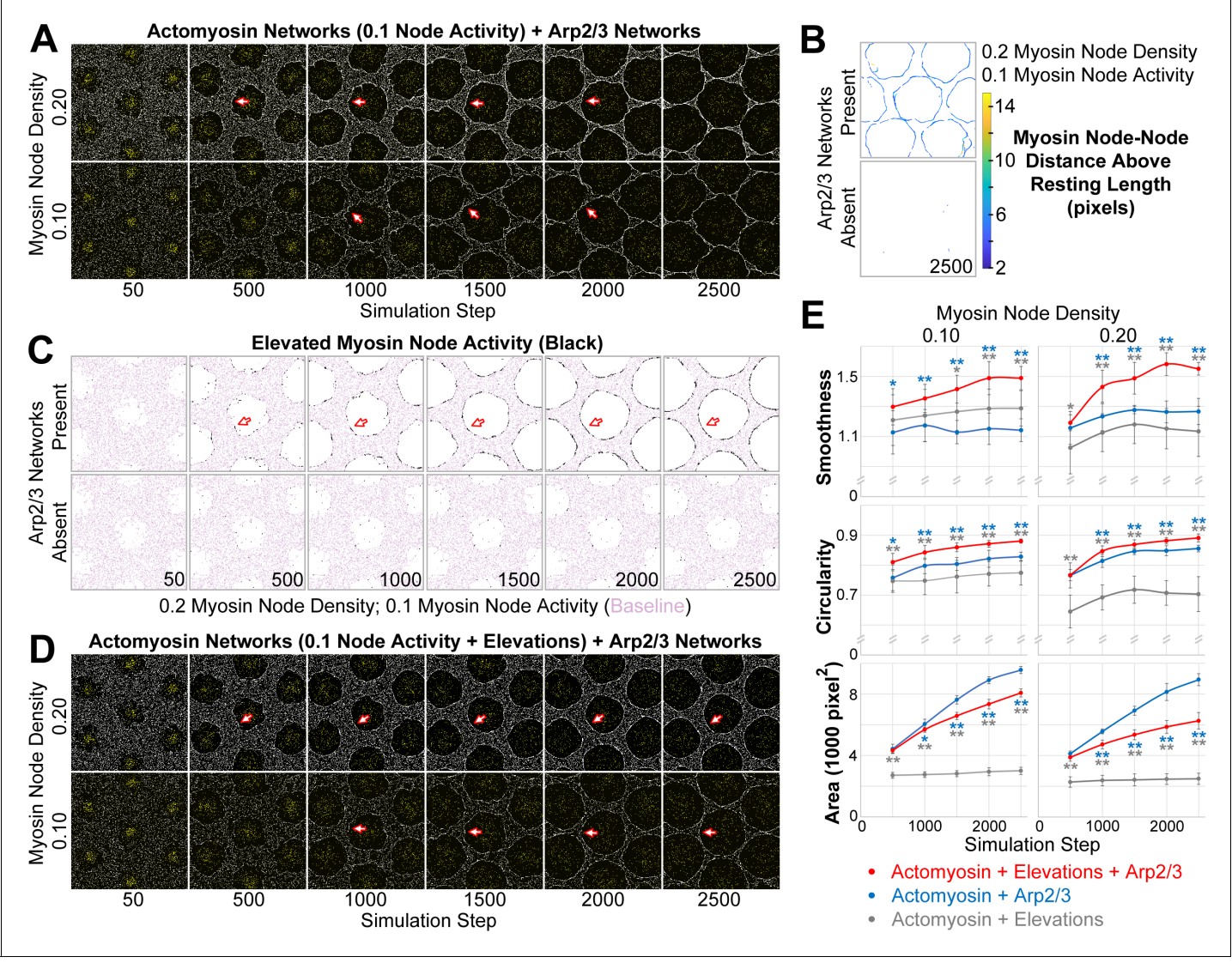

**Figure 4.** Emergence of a smooth interface from growth of an Arp2/3 network against a mechanosensitive actomyosin network. (**A–D**) Central views of configurations similar to **Figure 3B**. (**A**) Addition of growing Arp2/3 networks (yellow) to the clearances of actomyosin networks with low contractility (myosin nodes in white), led to clearance area increases (compare with **Figure 3D**) and accumulation of myosin nodes at the network-network interface (arrows), but the interfaces were wavy and the clearances became polygonal. These effects were documented at two myosin node densities (0.10 and 0.20). (**B**) Detection of myosin node-node distances greater than the resting length by two pixels or more (see color scale). The detections accumulated at the interface between the actomyosin network and growing Arp2/3 networks, and were rare in actomyosin networks without Arp2/3 networks. Shown at simulation step 2500. Representative of four replicates. Shown for myosin node density of 0.2, and also seen for myosin node density of 0.1 (**Figure 4—figure supplement 7A**). (**C**) Enrichment of myosin node activity elevations (black) above baseline (pink) at the interface of a mechanosensitive actomyosin network with growing Arp2/3 caps (arrows). The enrichment increased as the Arp2/3 network grew (compared simulation steps 500 to 2500), did not occur in other regions of the actomyosin network, and did not occur in the absence of growing Arp2/3 networks. Shown for myosin node density of 0.2, and also seen for myosin node density of 0.1 (**Figure 4—figure supplement 7B**). (**D**) Addition of growing Arp2/3 networks (yellow) to the clearances of actomyosin networks with low contractility and mechanosensitivity (myosin nodes in white), led to clearance area increases, accumulation of myosin nodes at the network-network interface (arrows), and interfaces that were smooth and circular. These effects were documented at two myosin node densities (0.10 and 0.20). Compared with the non-mechanosensitive actomyosin networks of (**A**), the mechanosensitive networks retained greater separation between clearances as the Arp2/3 networks grew (compare simulation steps 2500 in A and D). (**E**) Quantifications of the smoothness, circularity and area of the central clearance of simulations indicated. Means ± SD shown for eight simulations with distinct starting conditions. Blue asterisks compare mechanosensitive actomyosin networks plus Arp2/3 networks with non-mechanosensitive actomyosin networks plus Arp2/3 networks. Gray asterisks compare mechanosensitive actomyosin networks plus Arp2/3 networks with mechanosensitive actomyosin networks alone (example images of the latter in **Figure 4—figure supplement 7C**). Single asterisk; p<0.05: double asterisks; p<0.01.

The online version of this article includes the following figure supplement(s) for figure 4:

*Figure 4 continued on next page*

*Figure 4 continued*

**Figure supplement 1.** Sensitivity of interface smoothness and circularity to changes of search radius for establishing myosin node connections and of the maximum number of connections a myosin node can make.

**Figure supplement 2.** Sensitivity of interface smoothness and circularity to changes of maximum myosin node density per pixel and of internode spring force strength.

**Figure supplement 3.** Sensitivity of interface smoothness and circularity to changes of Arp2/3 network nucleation site density and of actin polymerization force.

**Figure supplement 4.** Sensitivity of interface smoothness and circularity to changes of Arp2/3 network branch timing and of plus end node loss probability.

**Figure supplement 5.** Sensitivity of interface smoothness and circularity to changes of Arp2/3 network plus end node loss timing and of maximum polymer length.

**Figure supplement 6.** Sensitivity of interface smoothness and circularity to changes of search radii distances controlling network-network interactions.

**Figure supplement 7.** Additional data related to *Figure 4*.

## Local activation of myosin nodes by pushing from plus end nodes leads to a smooth and circular interface

Although proper syncytial structure requires regulators that reduce cortical actomyosin activity (*Lee and Harris, 2013*; *Mason et al., 2016*; *Zhang et al., 2018*), our in vivo imaging revealed an Arp2/3-dependent accumulation of myosin at the smooth and circular interface between each cap and myosin border (*Figure 3A*, black arrows). Mechanical enhancement of actomyosin activity is evident in vivo (*Effler et al., 2006*; *Fernandez-Gonzalez et al., 2009*), and implicated mechanisms include stabilization of the myosin-actin interaction (*Kobb et al., 2017*; *Kovács et al., 2007*; *Yamashiro et al., 2018*), and alignment of actin filaments for optimal myosin activity (*Ennomani et al., 2016*). Thus, we hypothesized that the pushing forces of a growing cap would stabilize and/or align myosin nodes at the interface, and thereby locally increase myosin contractility within an otherwise weakly contractile network.

To incorporate mechanosensitive elevation of local myosin node activity into the simulation, the activity of a myosin node was increased 6-fold if it was displaced to a pixel already occupied by five or more myosin nodes (see Model Formulation). This approach did not explicitly link myosin node modification to the Arp2/3 networks, as the displacement of a myosin node for any reason could induce the activity elevation. However, comparison of myosin node activity elevation events with and without Arp2/3 networks revealed that the elevations occurred predominantly at interfaces between the actomyosin and Arp2/3 networks (*Figure 4C*, arrows), and that they required the Arp2/3 network (*Figure 4C*). These Arp2/3-dependent activations of interface myosin in silico resembled the Arp2/3-dependent accumulations of interface myosin in vivo (*Figure 3A*).

To test how local, Arp2/3-dependent, myosin node activity elevations affected the interface between the actomyosin and Arp2/3 networks, we compared simulations with and without the activations. Strikingly, the elevations significantly enhanced the smoothness and circularity of the interfaces (*Figure 4D*), compared to simulations without the elevations (*Figure 4A*). The areas of the clearances were also reduced. Quantifications of clearance smoothness, circularity, and area showed the importance of a combination of Arp2/3 network growth and mechanosensitivity of the actomyosin network (*Figure 4E*). Sensitivity analyses showed that the smooth and circular interfaces arose upon inclusion of mechanosensitivity over a wide range of parameter values affecting the myosin network, the Arp2/3 network, and their interface (*Figure 4—figure supplements 1–6*). The overall organization of the myosin networks simulated with mechanosensitivity and Arp2/3 network growth (*Figure 4D*) closely resembled the myosin networks formed in the presence of Arp2/3 actin caps in wild-type embryos (*Figure 3A*). These analyses suggest that a smooth and circular interface can form from the growth of an Arp2/3 network against a weakly contractile actomyosin network that is mechanosensitive to local activation.

## The smooth and circular interface forms with the local induction and contraction of an actomyosin ring

Since smoothening and circularization of the interface coincided with local enrichment of myosin nodes activated by the pushing forces of centrifugal Arp2/3 network growth, we hypothesized that re-shaping of the interface occurred through formation and contraction of an actomyosin ring within

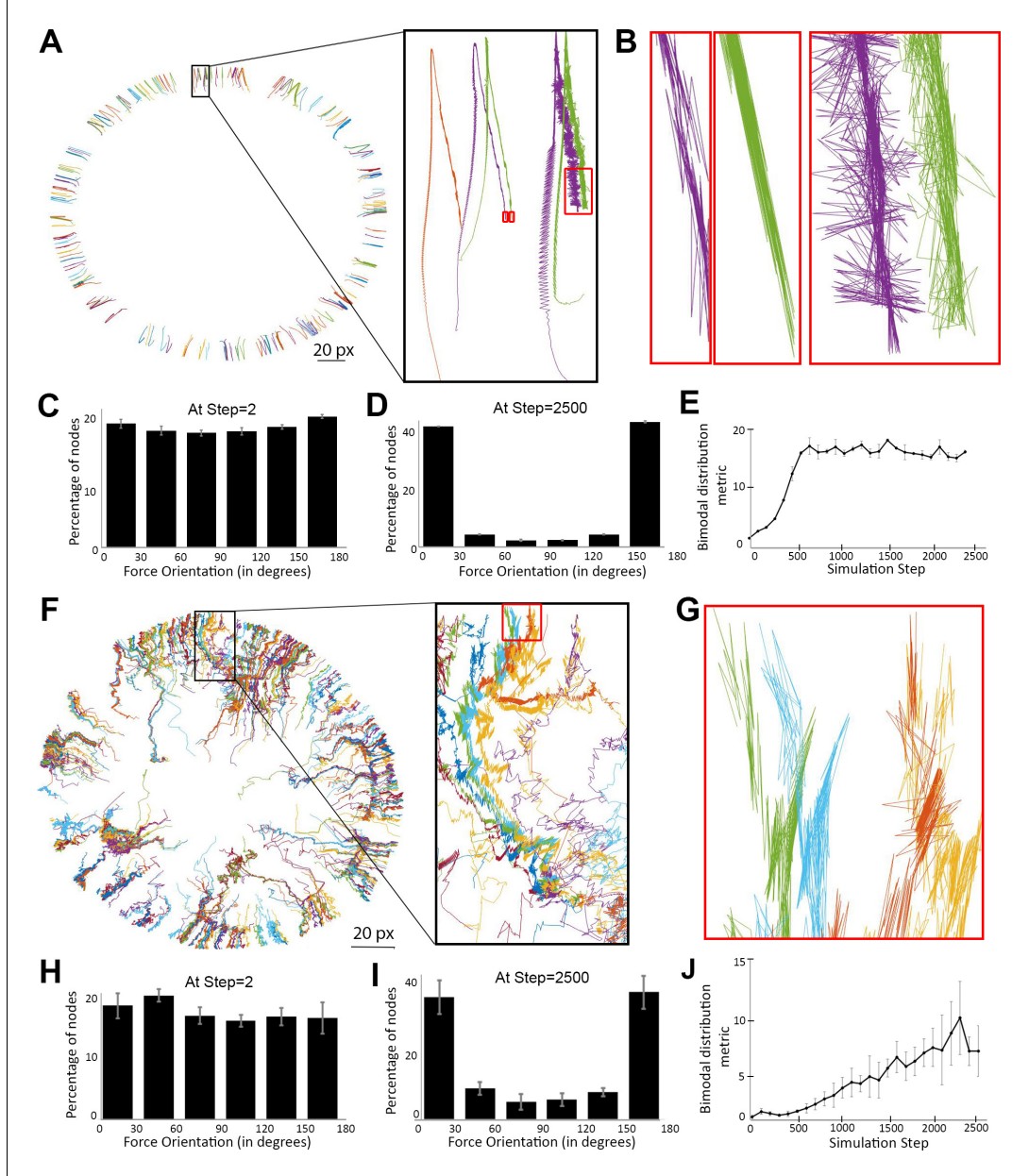

**Figure 5.** Myosin nodes activated by Arp2/3 network growth behave as a contractile ring during smoothening of the network–network interface. (**A**) Tracings of the displacements of individual myosin nodes selected around the inner circumference of a single actomyosin ring that started with a rough configuration, thinned, and constricted over 2500 simulation steps, similar to *Figure 2E*. Each line is the tracing of one node. The magnified inset (black box) shows the displacement of five nodes. The nodes move from the bottom half of the box to the top of the box as the ring thins, and then move back to the center of the box as the ring constricts. The red boxes indicate the final steps of the simulation. (**B**) Magnified views of the red boxes in (**A**) show back-and-forth displacements of the nodes, and that most of these displacements are closely aligned with the local radius of the circle (see circle center in (**A**)). (**C–D**) Histograms of node displacement angles relative to local radii of the circle at the beginning of the simulation when an even distribution occurred (**C**) and at the end of the simulation when the distribution became bimodal with most nodes in close alignment with the radii (**D**). Means ± SD shown for three simulations with distinct starting configurations. 7200–7400 nodes of the entire rings quantified per simulation. (**E**) Quantification of the simulations of (**C–D**) gaining bimodal distributions between steps 0 and 2500 by plotting the number of nodes oriented more parallel with the radii [those with angles of (0–30°) + (150–180°)] divided by the number of nodes oriented more perpendicular to the radii [those with angles of (60–120°)]. (**F**) Tracings of the displacements of individual activated myosin nodes selected around the boundary of a mechanosensitive actomyosin network at its interface with an expanding Arp2/3 network (similar to the central ring of the two-network analysis in *Figure 4D* with myosin node density of 0.2). Each line is the tracing of one node before and after elevation of its activity. The magnified inset (black box) shows the displacement of ~12 nodes. The nodes move from the bottom of the box to the top of the box as the network–network interface accumulates activated myosin nodes and smoothens. The red box indicates the final steps of the simulation. (**G**) Magnified view of the red box in (**F**) shows back-and-forth

*Figure 5 continued on next page*

*Figure 5 continued*

displacements of the nodes, and that most of these displacements are closely aligned with the local radius of the circle (see circle center in (F)). (H–I) Histograms of node displacement angles relative to local radii of the circle at the beginning of the simulation when an even distribution occurred (H) and at the end of the simulation when the distribution became bimodal with most nodes in close alignment with the radii (I). Means ± SD shown for three simulations with distinct starting configurations. 186–314 nodes quantified per simulation. (J) Quantification of the simulations of (H–I) gaining bimodal distributions between steps 0 and 2500, calculating the bimodal distribution metric as in (E).

an otherwise weakly contractile actomyosin network. To test this idea, we first examined the behavior of an isolated single ring of myosin nodes with uniform activity equal to the activity induced by Arp2/3 network growth in the two-network simulations. The isolated network was initiated as a thick, rough-edged ring of randomly positioned myosin nodes containing a central clearance instead of an Arp2/3 network (similar to the single ring in *Figure 2E*). Since the movement of a node is directly linked to the net forces applied to it by surrounding nodes, we traced node movements to map forces within the network. *Figure 5A* traces the movements of myosin nodes at the inner boundary of the rough ring. Consistent with our single ring analyses in *Figure 2E*, the inner boundary nodes initially moved centrifugally as the ring thinned (*Figure 5A*), and then moved centripetally as the thin and smooth ring constricted (*Figure 5A*). To determine the directions of net forces experienced by all individual nodes during the process, we calculated angles of node displacement relative to the circular network's radii at each node's position at each simulation step. At the beginning of the simulation, a broad distribution of node displacement angles was observed (*Figure 5C*), but during the transition from thinning to constricting, the distribution became biased toward 0° and 180° relative to local radii of the circular network (*Figure 5D–E*). To understand this bimodal distribution, we examined individual node displacements at full temporal and spatial resolution during the final stages of the simulation. Strikingly, nodes mostly displayed back-and-forth displacements closely aligned with the radii (*Figure 5B*). The angles of these oscillations are partly what would be expected for centripetal normal forces of a contractile circle, but with additional centrifugal counter-forces arising from the network's interconnected nodes. Although the in vivo relevance of these extremely rapid oscillations is unclear, the oriented oscillations provided a signature of contractile actomyosin ring formation in our simulations.

Next, we tested if the myosin nodes activated by Arp2/3 network growth in the two-network simulations displayed this signature of a contractile actomyosin ring. A subset of activated myosin nodes was selected and the positions of individual nodes were traced before and after activity elevation. Arp2/3 network growth displaced the myosin nodes centrifugally throughout the simulation, including stages of elevated myosin node activity and when the network-network interface became smooth and circular (*Figure 5F*). To map the directions of net forces applied to individual nodes, we determined angles of node displacement relative to radii from the clearance center throughout the simulation. Initially, a broad distribution of local displacement angles occurred (*Figure 5H*). In contrast, the phase of smoothening and circularization was accompanied by a bimodal distribution dominated by angles near 0° and 180° (*Figure 5I–J*). Full resolution tracing of nodes during the second phase showed that these angles were due to back-and-forth displacements that mainly had close alignment with local radii but occasionally occurred at other angles (*Figure 5G*). Overall, we conclude that the mechanical elevation of myosin node activity by the pushing forces of Arp2/3 network growth produces a contractile actomyosin ring around the network-network interface, and that contractility of this ring smoothens and circularizes the interface.

## Discussion

With relatively simple design principles, our node-based simulations of actomyosin and Arp2/3 networks individually mimicked the mesoscale behaviors of their respective networks reconstituted in vitro. The node-based structure common to each simulated network allowed straight-forward construction of two-network simulations that combined the actomyosin and Arp2/3 networks as segregated materials able to impact each other laterally through local interactions between their constituent nodes. Our simulations revealed that a myosin network with relatively low contractility is insufficient for generating smooth boundaries, and abnormally persistent rough boundaries mimicked those of myosin networks in embryos depleted of Arp3. In silico, the simple addition of

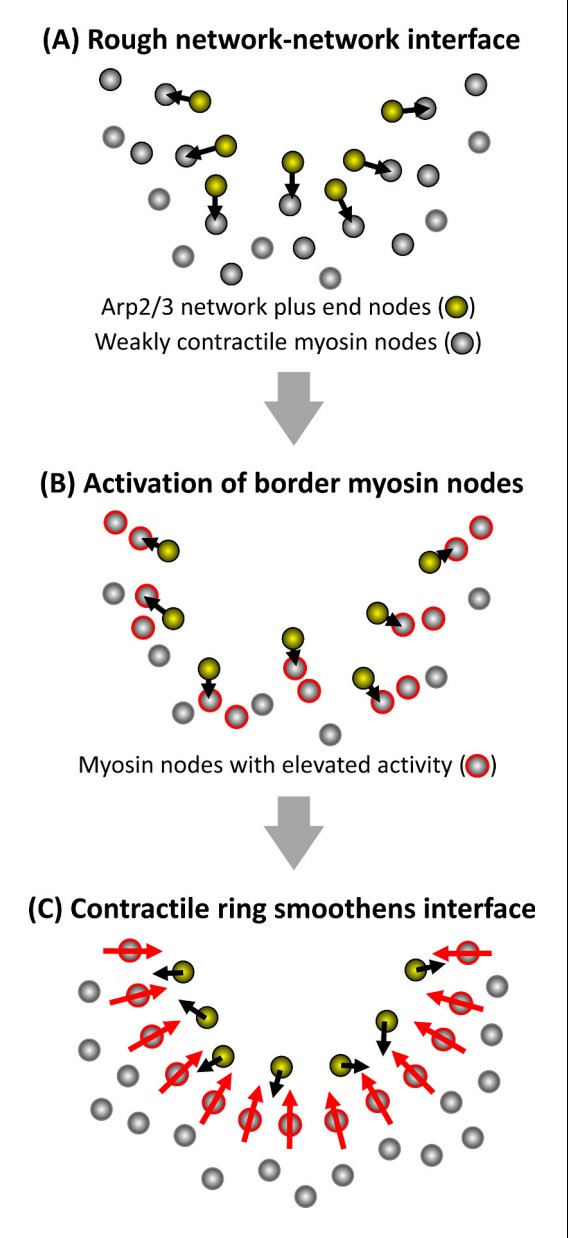

**Figure 6.** Schematic model of how a smooth interface forms from growth of an Arp2/3 network against a mechanosensitive actomyosin network. Half of a clearance is shown at three stages of the rough-to-smooth transition of the network-network interface. (**A**) Arp2/3 network plus end nodes (yellow circles) polymerize to the rough boundary of weakly contractile myosin nodes (gray circles). (**B**) The plus end nodes apply forces to interface myosin nodes which gain activity (gray circles with red outlines). (**C**) Activated myosin nodes form a local contractile ring (bottom half of ring shown) which smoothens the network-network interface.

growing Arp2/3 networks resulted in wavy, non-circular interfaces. However, local activation of an otherwise weakly contractile actomyosin network by the pushing forces of Arp2/3 network growth resulted in smooth, circular interfaces resembling those of wild-type embryos. Indicating robustness of the simulations, the effects of Arp2/3 network growth occurred over a substantial range of parameter values affecting the actomyosin network, the Arp2/3 network, and the network-network interface. Arp2/3-induced myosin node activations occurred around the initially rough network-network interface, and the activated myosin nodes locally formed a contractile ring to smoothen and circularize the interface (see schematic model in *Figure 6*).

The formation of an extended, smooth interface across a 2D plane is not trivial. Tensile forces can align polymers, but the two ends of the network must be anchored. Since a network can anchor to itself, as in the case of a cytokinetic ring (*Schwayer et al., 2016*), intra-network anchorage may explain the formation of smooth boundaries within a multiply fenestrated myosin network with high contractility (*Figure 3C*). However, low contractility networks cannot form smooth boundaries on their own. Low contractility is important for maintaining the syncytial state of the early *Drosophila* embryo (*Lee and Harris, 2013*; *Mason et al., 2016*; *Zhang et al., 2018*), arrays of podosomes at the base of invasive mammalian cells (*Rafiq et al., 2017*), and the polar body bud during meiosis II arrest of mouse oocytes (*Yi and Li, 2012*). Widespread contractility of a synthetic material would also hinder fabrication of a multi-component structure. Our simulations indicate that disruptive effects of widespread contractility can be avoided by the local induction of contractility in one material in response to pushing forces from a neighboring material, and that this induction can produce a contractile ring to smoothen the interface between such materials. In the *Drosophila* embryo, the smooth and circular interface demarcates where the plasma membrane folds to form a dome-like compartment. Without sufficient growth of the actin cap, the interface remains rough and membrane folding fails (*Figure 3A*; *Stevenson et al., 2002*; *Zallen et al., 2002*). With weakening of the myosin border, the actin cap grows with an irregular shape and the plasma membrane invagination is lopsided around the cap (*Zhang et al., 2018*).

Our simulations of network-network interactions at the surface of the early *Drosophila* embryo show how a smooth and circular structure can emerge from the local interactions of roughly distributed components. This example of refinement by mechanical self-organization is relevant to other cell biological processes, such as mammalian polar body budding (*Yi and Li, 2012*) and podosome protrusion (*van den Dries et al., 2019*). It is also relevant to the refinement of smart materials, including nanocomposites and metamaterials, after an initial printing step (*Begley et al., 2019*; *Holmes, 2019*). Biological composite materials can exhibit complex behaviors due to their multiple mechanical properties, and are thus top candidates in advanced manufacturing (*Eder et al., 2018*; *Wang et al., 2020*). Notably, reconstituted actin networks can be coated with inorganic materials to fabricate specific 3D structures (*Galland et al., 2013*). Although the in vitro reconstitution of Arp2/3 and actomyosin networks as neighboring domains remains a technical challenge, the patterned induction, mechanical self-organization, and inorganic coating of such composites holds exciting promise. Node-based modelling is also applicable to a range of materials, and could identify various roles for mechanical self-organization in device fabrication.

## Materials and methods

### Key resources table

| Reagent type (species) or resource | Designation | Source or reference | Identifiers | Additional information |
|---|---|---|---|---|
| Genetic reagent (*D. melanogaster*) | maternal-GAL4-VP16 | Mark Peifer | | |
| Genetic reagent (*D. melanogaster*) | UASp-Arp3-shRNA | Bloomington *Drosophila* Stock Center (BDSC) | BDSC #53972 RRID:BDSC_53972 | |
| Genetic reagent (*D. melanogaster*) | UASp-mCherry-shRNA | BDSC | BDSC #35785 RRID:BDSC_35785 | |
| Genetic reagent (*D. melanogaster*) | Zipper-GFP | Flytrap | BDSC #51564 RRID:BDSC_51564 | |
| Software, algorithm | MATLAB | MathWorks | | |
| Software, algorithm | Volocity | Quorum Technologies Inc | | |
| Software, algorithm | Image J | NIH | | |
| Software, algorithm | Excel | Microsoft | | |
| Software, algorithm | PowerPoint | Microsoft | | |
| Software, algorithm | Photoshop | Adobe | | |

## Model formulation

MATLAB code is available at: https://github.com/medha7575/sharma-et-al-ELIFE2021. Annotated PDFs of the code are provided as *Source code 1*, *2* and *3*.

Node-based models were created for the actomyosin network, the Arp2/3 network, and the interface between the two networks. Nodes change their positions synchronously during each simulation step. We assume a low Reynold's number, in which viscosity dominates (*Purcell, 1977*), and thus there is no inertia between steps. Unless otherwise stated, simulation rules were standardized across our analyses.

## Simulation of the actomyosin network

A 2D array of nodes simulated the properties of a contractile actomyosin network (*Banerjee et al., 2020*; *Lecuit et al., 2011*; *Mangione and Gould, 2019*). Each node represents a myosin mini-filament. Interconnecting actin filaments are not explicitly modeled, but would be cross-linked in anti-parallel orientations to allow plus-end directed myosin activity to pull nodes together. Each myosin motor domain binds an actin filament, uses ATP-derived energy to displace the filament, and then dissociates from the filament. Since a mini-filament contains multiple motor domains, this cycle can be repeated by separate motor domains, allowing processive displacement of an actin filament by a myosin mini-filament. The activity of a motor domain can be regulated, and the number of motor domains in a mini-filament can vary. We incorporated these properties into our model by assuming that actin-based connections between nodes are never lost, and that the pulling force between two connected myosin nodes is proportional to the product of an adjustable myosin activity at each node. Considering only two myosin nodes ($n_1$ and $n_2$) with equal myosin activities ($M_1$ and $M_2$), pulling between $n_1$ and $n_2$ would displace both nodes by the same distance toward each other. The force experienced by each node is defined by $K_{myosin}.(M_1.M_2)U$, where the unit vector, $U$, defines the direction of pulling force based on the relative positions of $n_1$ and $n_2$, and the coefficient $K_{myosin}$ assigns units of force to the myosin activity (*Equation 1*).

To generate an array of many myosin nodes, the nodes are distributed randomly over a 500 by 500 lattice of pixels. Nodes at the lattice boundary maintain their positions even when a force is applied on them (a fixed boundary condition). Each node is visualized as a single pixel. However, node connection angles, node connection distances, force angles, and node displacements are defined at 1/100th of a pixel. Since the cell cortex is 50–400 nm thick (*Chugh and Paluch, 2018*; *Salbreux et al., 2012*; *Svitkina, 2020*), the positioning of nodes above and below each other in 3D space is represented in the 2D simulation by allowing a maximum of 20 nodes to occupy the same pixel (with no information encoding the third dimension) (see *Table 1* for a full list of parameters). Incorporating this allowance avoided jamming of nodes into a non-dynamic state, but to mimic local steric hindrance within the cell cortex the upper limit to the number of nodes per pixel was implemented. If a node attempts to move to a fully occupied pixel, then the move is disallowed and the node maintains its original location leading to a configuration akin to a mini-filament stack (*Henson et al., 2017*; *Hu et al., 2017*).

After the myosin nodes are positioned across 2D space they are linked by permanent connections. Each node is randomly connected with a maximum of six other nodes within a search radius of 5 pixels (500 nm). This design is based on myosin mini-filaments (i) having 28–58 motor domains (*Billington et al., 2013*; *Niederman and Pollard, 1975*), which would not all be simultaneous bound to actin (discussed above), (ii) displaying motor domains at angles of up to 180° relative to each other (*Billington et al., 2013*; *Laplante et al., 2016*; *Niederman and Pollard, 1975*; *Sellers and Kachar, 1990*), and (iii) engaging actin filaments with lengths of ~500 nm for optimal contractility of cortical actomyosin networks (*Chugh et al., 2017*). If a connection is attempted with a saturated neighbor, then the attempt is aborted. If the node density is low, then a node may not maximize its connections in the available search radius. By considering pairwise relationships between nodes instead of modeling actin filaments, node-node pairings can cross each other in the 2D simulation, representing actin filaments crossing each other at different levels of 3D space.

Once an interconnected array of nodes is assembled, myosin activity is triggered to simulate network contraction. The pulling forces from 'n' connected neighboring myosin nodes on the 'm$^{th}$' myosin node are determined by:

$$\vec{F}_{m,\mathrm{myosin}} = K_{\mathrm{myosin}} \sum_{i=1}^{n} (M_m * M_i) \hat{U}_{mi} \tag{1}$$

where,

$\vec{F}_{m,\mathrm{myosin}}$: Net force due to inter-node pulling forces on the 'm$^{th}$' node.

$K_{\mathrm{myosin}}$: Coefficient of myosin force.

$M_m$: Myosin node activity level of the 'm$^{th}$' node.

$M_i$: Myosin node activity level of the 'i$^{th}$' connected node.

$\hat{U}_{mi}$: Unit vector pointing in the direction of the i$^{th}$ node from the m$^{th}$ node.

In addition, cross-linked actin networks display elastic responses to stretching but not compression (**Banerjee et al., 2020**; **Salbreux et al., 2012**). This asymmetric elasticity is incorporated by activating spring-like forces between the myosin nodes in response to stretching, but not to compression. The spring forces on the 'm$^{th}$' myosin node are determined by:

$$\vec{F}_{m,\mathrm{spring}} = K_{\mathrm{spring}} \sum_{i=1}^{n} (li - lo) \hat{U}_{mi} : \mathrm{if}\ li > lo$$
$$\vec{F}_{m,\mathrm{spring}} = 0 : \mathrm{otherwise} \tag{2}$$

where,

$\vec{F}_{m,spring}$: Net force due to spring-like forces on the 'm$^{th}$' node.

$K_{\mathrm{spring}}$: Coefficient of spring force $l_i$-$l_o$: Current length of connection – resting length of connection $l_o$: Mean length of all connections before a simulation starts.

$\hat{U}_{mi}$: Unit vector pointing in the direction of the i$^{th}$ node from the m$^{th}$ node.

A damping coefficient, $\xi$, of 0.1 nN.s.$\mu$m$^{-1}$ (**Belmonte et al., 2017**; **Matsuda et al., 2019**) determines the node displacement induced by the calculated net force. The damping coefficient is held constant throughout all simulations in this study. The displacement of the 'm$^{th}$' myosin node is determined by:

$$\Delta s_m = \left( \vec{F}_{m,\mathrm{myosin}} + \vec{F}_{m,spring} \right).\Delta t \xi^{-1} \tag{3}$$

where,

$\Delta s_m$: Displacement of the 'm$^{th}$' node.

$\Delta t$: Time difference (one simulation step).

$\xi$: Damping coefficient.

Using the pre-existing position and the net displacement of the 'm$^{th}$' myosin node, its new position is calculated after every simulation step by:

$$\mathrm{New\,Position} = \mathrm{Old\,Position} + \Delta s_m \tag{4}$$

## Simulation of the Arp2/3 network

Like the actomyosin model, the Arp2/3 network was simulated as a 2D array of nodes. In the Arp2/3 actin model, however, actin plus end nodes move away from stationary nucleation points representing Arp2/3 nucleation sites, and the overall array expands. The filaments themselves are not explicitly modelled. Since the Arp2/3 complex nucleates actin polymerization (**Pollard, 2016**; **Svitkina, 2018**) and centrosomes act as spatial landmarks for the induction of Arp2/3-enriched actin caps in *Drosophila* (**Raff and Glover, 1989**; **Stevenson et al., 2001**), an Arp2/3 network is initiated by randomly positioning the nucleation points in a dense inner circle (20 pixel radius with 10% of pixels containing a nucleation point) and a sparse, larger and overlapping circle (50 pixel radius with 1% of pixels containing a nucleation point). The outer circle of sparse nucleation sites is expected from graded diminishment of a centrosome-based signal, and was needed to generate an Arp2/3 network that grew with an irregular shape, as occurs in vivo (**Zhang et al., 2018**). Actin polymerization is simulated by displacing a plus end node away from its corresponding nucleation point. The nucleation point is only used to define the filament position, length, and angle, in combination with the positional information of the plus end node. The polymerization and associated pushing force is modeled by the plus end node only. At the simulation start, plus end nodes move from their nucleation

points in random directions. During the simulation, the nucleation points maintain their position, assuming a stabilizing effect of the larger network.

The polymerization force of a plus end at each simulation step is determined by:

$$\vec{F}_{\text{poly}} = \text{K}_{\text{poly}} * \hat{\text{U}} \tag{5}$$

where,

K$_{\text{poly}}$: Actin polymerization force coefficient.

$\hat{\text{U}}$: Unit vector pointing in the direction of the plus end node from the nucleation point.

Thus, the plus end node will move to a new position given by:

$$\text{New Position} = \text{Old Position} + \Delta\mathbf{s}_{\text{poly}} \tag{6}$$

where,

$$\Delta\text{s}_{\text{poly}} = \vec{F}_{\text{poly}}.\Delta\text{t}.\xi^{-1} \tag{7}$$

To form a dendritic network, the Arp2/3 complex nucleates growth of a new actin polymer at a 70˚ angle relative to a preexisting polymer (*Pollard, 2016*; *Svitkina, 2018*). In vitro, branching can occur from either side of the mother filament, anywhere along the mother filament (with a preference toward the barbed end), and the mother and daughter filaments polymerize at the same rate (*Amann and Pollard, 2001*). To implement branched polymerization from an existing filament, the age of the plus end node is monitored, and after nine simulation steps a new nucleation point is created halfway between the pre-existing nucleation point and the plus end node. From the new nucleation point, a new plus end node extends with an angle of ±70˚ (randomized) relative to the mother filament and its plus end node. One actin filament is limited to forming one branch.

Two major effects regulate the growth and structure of Arp2/3 networks. Capping of actin plus ends promotes shorter filament lengths that resist buckling and exert more effective pushing forces, whereas depolymerization of older filaments by actin depolymerization factors allows network turnover (*Banerjee et al., 2020*; *Pollard, 2016*; *Pollard and Borisy, 2003*; *Svitkina, 2018*). To simulate inhibition of polymerization by a capping protein, the polymerization force of a plus end node is set to zero after it extends 20 pixels from its nucleation point. To model actin filament depolymerization, a plus end node is removed with a probability of 0.7 after it exists for 20 simulation steps. After the deletion of a plus end node, the previously coupled nucleation point becomes dormant, but a branch formed between the pair remains in place, can continue growing, and can induce its own branch. By linking depolymerization to the lifetime of a filament, filament turnover occurs even when growth is prevented by an obstruction. At the start of the simulation, it was necessary to randomly assign a range of initial filament lengths and ages to prevent filament capping and removal from occurring in an unnaturally rhythmic way.

The Arp2/3 network simulation is considered to be a broad 2D array with a 50–400 nm thickness in 3D (*Chugh and Paluch, 2018*; *Salbreux et al., 2012*; *Svitkina, 2020*). Like the actomyosin model, this organization is implemented by positioning nodes on an off-lattice model to avoid steric hindrance, with close positioning of two nodes in 2D representing their occupancy of different levels of the same vertical column in 3D. Unlike the actomyosin network, there was no need to create an upper limit to the number of nodes at a pixel because nodes of the Arp2/3 network continually expand away from each other.

## Simulation of the network-network interface

To enable study of the lateral interaction between two segregated cytoskeletal networks, we combined the actomyosin and Arp2/3 networks into a single lattice. Although actomyosin networks and Arp2/3 networks form segregated domains with close neighbor relationships during *Drosophila* embryo pseudocleavage (*Foe et al., 2000*; *Zhang et al., 2018*), mammalian oocytes polar body budding (*Yi and Li, 2012*), and mammalian cell podosome formation (*van den Dries et al., 2019*), we are unaware of in vivo or in vitro studies addressing the structural properties of the lateral interface between the neighboring networks. In lieu of experimental data, we reasoned that displacement of a node at the interface would be dictated by a balance of local forces on the node from nodes of its own network and from nodes of the neighboring network. In the two-network model,

the constituent actomyosin and Arp2/3 networks have the same individual properties described in the previous sections, but all nodes of each network additionally perform two searches that indicate if a node is in proximity to a network-network interface. If so, the searches trigger mechanisms of internetwork displacement and segregation. Each node searches for nodes of the other network in a five-pixel radius. If a node of the distinct network is detected, then displacement of the searching node is dictated by both the rules of its own network and by effects of the neighboring network (as described in the following paragraphs). However, a potential displacement can be overridden. To maintain the network segregation apparent in vivo (*Foe et al., 2000*; *Mavrakis et al., 2009*; *Zhang et al., 2018*), each node searches a two-pixel radius around its potential displacement position for nodes of the other network. If a node of the other network is detected in this area, then the searching node aborts the move and maintains its position during the simulation step. Although the mechanistic basis of the in vivo network segregation is unclear, we assume involvement of steric hindrance and immiscibility of branched networks and bundled networks.

The displacement of an interface myosin node is determined by the aforementioned rules of the actomyosin network, and additionally by the pushing forces of actin plus end nodes within the five-pixel search radius. The net effect of the nearby plus end nodes on the new position of the myosin node is based upon the degree to which each expected plus end node displacement is directed toward the myosin node. The component of a plus end node polymerization force directed toward the myosin node is determined by $\phi$: the angle between the spatial vector joining the plus end node to the myosin node and the displacement vector calculated for the plus end node. If $\cos(\phi) < 0$, then the plus end node is about to move away from the myosin node, and would not push the myosin node. If $\cos(\phi) > 0$, then the plus end node displacement is directed toward the myosin node to a certain degree, and the component of the pushing force directed toward the myosin node is determined by $\cos(\phi)$. The force on the myosin node by the cumulative effects of 'n' nearby plus end nodes is calculated by:

$$\vec{F}_{m,actin} = \sum_{i=1}^{n} \vec{F}_{poly} \cos(\phi_i) : for \cos(\phi) > 0$$
$$\vec{F}_{m,actin} = 0, \text{otherwise}$$

(8)

The new position of the myosin node is determined by forces from both the actomyosin network and the nearby plus end nodes:

$$\Delta \mathbf{s}_{m,net} = \left( \vec{F}_{m,actin} + \vec{F}_{m,myosin} + \vec{F}_{m,spring} \right) . \Delta t \xi^{-1}$$
$$\text{New Position} = \text{Old Position} + \Delta \mathbf{s}_{m,net}$$

(9)

The myosin node moves to the new position, unless a plus end node is within a 2-pixel radius of the position, or if the position has the maximum myosin nodes allowed. If the move is prevented, then the myosin node maintains its old position.

The displacement of an interface actin plus end node is determined by the aforementioned rules of the Arp2/3 network, and by additional forces exerted by myosin nodes within the plus end node's five-pixel sensing radius. The networks are assumed to lack an interconnection for conveying pulling forces, and thus the myosin nodes can only impact the plus end node through pushing forces associated with myosin node displacement. At each simulation step, the contribution of the myosin-based pushing force to the displacement of the plus end node is based on the anticipated displacement of the myosin node directed toward the plus end node. The forces responsible for the myosin node displacement arise from the pulling and elastic forces between it and its associated myosin nodes. The component of this displacement directed toward the plus end node is determined by $\phi$: the angle between the spatial vector joining the myosin node to the plus end node and the displacement vector calculated for the myosin node. If $\cos(\phi) < 0$ then the myosin node is about to be displaced away from the plus end node, and would thus have no pushing effect on the plus end node. If $\cos(\phi) > 0$ then the myosin node displacement is directed toward the plus end node to a certain degree, and the component of the pushing force directed toward the plus end node is determined by $\cos(\phi)$. Considering 'n' pixels in the sensing radius of a plus end node have myosin nodes with $\cos(\phi) > 0$, the forces on the plus end node are summed as:

$$\vec{F}_{a,myo} = \sum_{i=1}^{n} N_i . \left( \vec{F}_{m,\text{myosin}} + \vec{F}_{m,spring} \right)_i . \cos(\phi_i) ; for \cos(\phi) > 0$$
$$\vec{F}_{a,myo} = 0, \text{otherwise}$$
(10)

where $N_i$ represents the number of myosin nodes on pixel i.

The new position of the plus end node is determined by both its polymerization force and the forces from nearby myosin nodes:

$$\Delta \mathbf{s}_{\text{actin}} = \left( \vec{F}_{a,myo} + \vec{F}_{poly} \right) . \Delta t \xi^{-1}$$
$$\text{New Position} = \text{Old Position} + \Delta \mathbf{s}_{\text{actin}}$$
(11)

The plus end node moves to the new position, unless a myosin node is within a two-pixel radius of the position. If a myosin node exists in this radius, then the plus end node maintains its old position.

## Software

Simulation codes were written and run using MATLAB (MathWorks). Annotated codes are provided (Supplemental files). MATLAB and Image J (NIH) were used for quantifications. Excel (Microsoft) was used for graphing means ± SD, and for determining p values with T-tests of samples with unequal variance and two-tailed distributions. PowerPoint (Microsoft) was used for schematics. Photoshop (Adobe) was used for figure preparation.

## Quantifications

### Arp2/3 network area

To quantify the area of growing Arp 2/3 networks with and without a wall (*Figure 1I*), the outermost plus end nodes of the network were connected manually using the polygon selection tool in Image J, and then the area of the polygon was calculated by Image J.

### Inner clearance area

To quantify inner clearance areas in single ring simulations (*Figure 2G*), a custom tool script was written in MATLAB. For each simulation step, the tool used a swarm-based approach to detect the boundary of the clearance and then calculated the area of the clearance. Each input image of the myosin nodes was preprocessed by conversion to grayscale, application of a 2x2 pixel median filter, and conversion to binary black and white. The tool then released 180 swarms from a central position and the swarms moved outward along straight trajectories each separated by two degrees (spanning 360 degrees in total). As each swarm moved, it collected pixel intensity values and coordinate values. The boundary pixel was determined as the pixel where the detected intensity became greater than the mean intensity. However, two occasional errors occurred: (i) swarms passed through the region of myosin nodes without encountering a node or (ii) stray myosin nodes were detected. To remove these errors, the data was divided into six 60 degree segments, the median of the clearance boundary distance measurements was calculated for each segment, and detections of the boundary beyond ±5 pixels of the median were then deleted. The detection of the clearance boundary by the swarms was first used to correct a manually estimated center of the clearance. To maximize the accuracy of the clearance area calculation, each clearance boundary was determined 50 times with swarms emanating from randomized central positions in each case. Clearance area was calculated from the detected boundary coordinates using the surveyor's formula (*Braden, 1986*).

### Interface smoothness, circularity and area

For quantifications of interface properties in the multi-ring simulations (*Figure 4E*), the boundary of the central clearance of myosin was first manually traced in Image J using the polygon selection tool and the following steps: (1) from a starting boundary myosin node subsequent boundary myosin nodes were traced around the clearance circumference until the starting node was reached again; (2) if two subsequent nodes were the same distance away, the node closer to the clearance center was chosen; and (3) after tracing the full boundary, it was corrected by comparing the tracing with the distribution of boundary myosin nodes at earlier simulation steps. The traced polygon was

subjected a two pixel-radius Gaussian blur and was then converted into a binary black and white mask. The area and circularity of the mask was calculated in Image J, with circularity having a maximum value of one. To calculate the smoothness parameter, we first employed the swarms tool to determine the coordinates of the mask boundary using 180 swarms. The swarm-derived data was divided into six segments of 60 degrees each. For each segment, the standard deviation (σ) of the derivatives of the radii was calculated. The derivatives of the radii were calculated by dividing the difference between consecutive swarm radii by half the angle between the two points. Smoothness, $S$, of the entire boundary of the central clearance was calculated as the inverse of the mean of the standard deviations, σ, over the six segments, as follows:

$$S = \left( \frac{\sum_{i=1}^{6} \sigma_i}{6} \right)^{-1} \tag{12}$$

The calculated smoothness approaches infinity for perfectly smooth shapes, but the pixelated images generated by MATLAB are not perfectly smooth, and we calculated $S$ close to ∼nine from MATLAB-generated circular plots.

### *Drosophila* work

Animals were maintained under standard conditions. True breeding stocks were maintained at room temperature, 18°C or 25°C on fly food provided by a central University of Toronto kitchen operated by H. Lipshitz. Embryos were collected on plates of apple juice agar (25 g agar, 250 ml store-bought apple juice, 12.5 g store-bought white sugar, 10 ml 10% Tegosept (in ethanol), plus dH$_2$O to 1000 ml) plates at 25°C after 2–3 days of caged adult feeding on dabs of store-bought baker's yeast with daily plate changes. Adults were caged for embryo collection within 1 week of pupal hatching, and no health issues were noticed. Embryo sexes were not determined and embryo populations with a specific genotype of interest displayed relatively normally distributed phenotypes suggesting no detectable sex contribution.

UAS constructs for Arp3 RNAi (UASp-Arp3-shRNA; Bloomington *Drosophila* Stock Center (BDSC) #53972) and control RNAi (UASp-mCherry-shRNA; BDSC #35785) were expressed maternally using maternal-GAL4-VP16 (gift of Mark Peifer) and myosin localization was assessed in offspring using a GFP insertion into the *zipper* (*zip*) gene locus (Zip-GFP; Flytrap #51564). Standard *Drosophila* genetics synthesized the maternal genotypes used: maternal-GAL4-VP16 /+; Zip-GFP Trap/UASp-Arp3-shRNA and maternal-GAL4-VP16 /+; Zip-GFP Trap/+; UASp-mCh-shRNA/+.

For live embryo imaging, dechorionated embryos were glued to a coverslip using tape adhesive dissolved in heptane and mounted in halocarbon oil (series 700; Halocarbon Products). The coverslip, with the embryos facing up, was set into the bottom of a glass bottom culture dish with its original coverslip removed. Images were collected with a spinning-disk confocal system (Quorum Technologies Inc) at RT with a 63x Plan Apochromat NA 1.4 objective (Carl Zeiss, Inc), a piezo top plate, an EM CCD camera (Hamamatsu Photonics), and Volocity software (Quorum Technologies Inc). Z stacks had 300 nm step sizes. Images were analyzed with Volocity software and ImageJ (NIH).

## Acknowledgements

We thank Rudi Winklbauer and Rodrigo-Fernandez-Gonzalez for their comments on the manuscript. This work was supported by Natural Sciences and Engineering Research Council of Canada Discovery grant RGPIN-2016–05617 to T Harris. M Sharma received a Connaught International Scholarship from the University of Toronto.

## Additional information

### Funding

| Funder | Grant reference number | Author |
| --- | --- | --- |
| Natural Sciences and Engi- | RGPIN-2016-05617 | Tony J C Harris |

neering Research Council of
Canada

The funders had no role in study design, data collection and interpretation, or the
decision to submit the work for publication.

## Author contributions

Medha Sharma, Conceptualization, Formal analysis, Investigation, Methodology, Writing - original
draft; Tao Jiang, Investigation; Zi Chen Jiang, Carlos E Moguel-Lehmer, Methodology; Tony J C Harris, Conceptualization, Resources, Supervision, Funding acquisition, Project administration, Writing - review and editing

## Author ORCIDs

Tony J C Harris (iD) https://orcid.org/0000-0002-0798-970X

## Decision letter and Author response

Decision letter https://doi.org/10.7554/eLife.66929.sa1
Author response https://doi.org/10.7554/eLife.66929.sa2

## Additional files

### Supplementary files

• Source code 1. Annotated code for the model of the actomyosin network alone. MATLAB code is
available at: https://github.com/medha7575/sharma-et-al-ELIFE2021.

• Source code 2. Annotated code for the model of the Arp2/3 actin network alone. MATLAB code is
available at: https://github.com/medha7575/sharma-et-al-ELIFE2021.

• Source code 3. Annotated code for the model of the two networks combined. MATLAB code is
available at: https://github.com/medha7575/sharma-et-al-ELIFE2021.

• Transparent reporting form

### Data availability

All data generated or analysed during this study are included in the manuscript and supporting files.

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
