## [Decision Letter]

**Acceptance summary:**

This manuscript presents computer simulations that demonstrate how smooth boundaries can be created between a zone of active expansion and a second zone of active constriction that surrounds it. This work is of interest to scientists working in the field of cytoskeletal organization, mechanobiology, and self-organizing active matter.

**Decision letter after peer review:**

Thank you for submitting your article "Emergence of a smooth interface from growth of a dendritic network against a mechanosensitive contractile material" for consideration by *eLife*. Your article has been reviewed by 2 peer reviewers, one of whom is a member of our Board of Reviewing Editors, and the evaluation has been overseen by Anna Akhmanova as the Senior Editor. The reviewers have opted to remain anonymous.

Essential revisions:

1) Provide a physical (or mechanistic) explanation for the observed smoothening behavior.

2) Test how sensitive the main results are to initial conditions, and to different parameter values by performing a sensitivity analysis (especially on the wild-type like cases).

3) Describe the model a bit more specifically at the beginning of the result section so the reader has a general sense of how it was implemented.

4) Using physically relevant units in the text and the table would make it easier to follow and potentially more impactful. Or the authors should explain if there is a good reason to keep those non-physical units.

5) Please provide Matlab files at resubmission.

*Reviewer #1:*

The authors attempt to understand the basis of smoothing interfaces between Arp2/3-based actin networks and actomyosin networks in the syncytial *Drosophila* embryo during pseudo-cleavage. This is an interesting question in the context of how actin networks are organized. To understand interface smoothing, the authors perform "node-based" simulations of the two actin networks. The simulation methods of the two networks follows previously published methods and are validated through comparisons with previous work. For the interface between these networks, the authors invoke some ad hoc rules that are poorly justified form a physical point of view. It is not clear, how much their results depend on this choice of interfacial dynamics. Whereas the simulation results are quantitatively analyzed, there is essentially no quantification of the experiments, and the comparison between experiment and theory remains qualitative. Eventually, the authors propose a set of ingredients that are sufficient to yield smooth interfaces in their simulations. However, a thorough understanding in terms of the physical mechanism underlying the simulation results remains elusive. For these reasons, in its current state, this works seems of limited use for the community.

*Reviewer #2:*

In this manuscript, Sharma et al., aim to better understand the mechanical mechanisms underlying the organization of the syncytium, the cortical meshworks of actin and myosin under the surface of the fly embryo during early embryo development. This meshwork is composed of two networks that self-organize into circular caps containing branched actin filaments and surrounded by an acto-myosin constrictive network. Using a model based on nodes that follow simple mechanical rules, the authors demonstrate that one can start with roughly defined zones where each meshwork is defined, and only in certain conditions these zones will become clearly defined with a smooth border as seen in the fly embryo. They show that clear and smooth boundaries appear only when a branched actin meshwork expands in the caps and when there the surrounding acto-myosin network exerts enough contractile forces especially at the interface between both meshworks.

Strengths:

The modeling of the actin and myosin meshworks is performed using phenomelogical rules at a meso-scale, which allow the authors to focus on general mechanical properties of the meshworks and limit the number of free parameters.

The authors nicely demonstrate what mechanisms cannot lead to smooth boundaries and which one can.

The authors systematically compare their simulations with experimental data in wild-type and in knock-down flies that have been published before or that they generate for this study.

Weaknesses:

The parameters are discussed in the text in arbitrary non-physical units (e.g. pixels, steps, etc.). It is sometimes difficult to follow whether the values used are realistic or not. That said the parameter table gives some equivalence but this prevents a smooth read.

Some of the parameters used in all the simulations are set to certain values but those values are often not discussed, and it is unclear to which extent the results of the simulations are sensitive to those parameters.

The code is provided only as a pdf file which prevents the reader to run the simulations themselves.

The authors could test how sensitive their main results are to initial conditions, and to different parameter values by performing a sensitivity analysis (especially on the wild-type like cases).

Please provide Matlab files at resubmission.

Things that could be clarified:

The authors could explain a bit more how the initial rough patterns of Arp2/3 and acto-myosin zones are generated in the embryo. Is it signaling? Could it be self-organized from a random initial organization by the mechanics too?

They could describe the model a bit more specifically at the beginning of the result section so the reader has a general sense of how it was implemented.

Using physically relevant units in the text and the table would make it easier to follow and potentially more impactful. Or the authors should explain if there is a good reason to keep those non-physical units.

---

## [Author Response]

Essential revisions:1) Provide a physical (or mechanistic) explanation for the observed smoothening behavior.

It was important to flesh out the mechanism involved. Our first submission showed that centrifugal expansion of a rough Arp2/3 network against a surrounding and rough actomyosin network is insufficient for forming a smooth interface between them, but that Arp2/3 growth did displace myosin nodes and increase strain between them. Based on mechanosensitive strengthening of actomyosin networks reported in the literature, we allowed myosin nodes to increase their activity with a specific degree of displacement. As a result, myosin nodes gained activity in a ring around the interface of the Arp2/3 network and the surrounding actomyosin network, specifically in response to Arp2/3 network growth, and the interface became smooth and circular. Our assumption/hypothesis was that formation of the smooth and circular interface occurred through contractility of the induced actomyosin ring. However, we agree that we only identified ingredients responsible for forming the smooth and circular interface, and that a physical explanation was lacking. To test the idea that the induced myosin nodes form a contractile ring within an otherwise weakly contractile actomyosin network, we compared simulations of an isolated actomyosin ring with the behaviours of myosin nodes activated by Arp2/3 in an otherwise low contractility network. All nodes of the isolated ring had activity equal to that induced by Arp2/3 in the two-network simulations and were initially arranged as a wide and rough-edged ring. With contractile activity of the nodes, the ring first thinned and smoothened and then the ring constricted centripetally. In the initial rough configuration, the net forces applied to individual nodes occurred at a broad range of angles, but as the ring thinned and constricted, these angles became locally aligned with radii of the contractile ring. We considered this alignment to be a signature of a contractile ring in our simulations. Myosin nodes activated around the circumference of a growing Arp2/3 network displayed a similar signature. Thus, we conclude that the smoothening behaviour is based upon the local induction and activity of a contractile actomyosin ring.

These observations are described in detail in an added Results section (starting at line 350) and shown in added Figure 5. We also added a schematic model as Figure 6 to explain the steps of the smoothening process, and refer to this figure in the first paragraph of the Discussion section (starting at line 410).

2) Test how sensitive the main results are to initial conditions, and to different parameter values by performing a sensitivity analysis (especially on the wild-type like cases).

Sensitivity analyses were also important to add. We performed these analyses to confirm the robustness of our two main findings: (1) that centrifugal expansion of a rough Arp2/3 network against a surrounding and rough actomyosin network is insufficient for forming a smooth interface between them, and (2) that the smoothening occurs through the pushing forces of the Arp2/3 network elevating myosin activity around the interface to form a contractile ring. We conducted these analyses by varying the values of all major parameters of the actomyosin network, the Arp2/3 network, and the network-network interface. Parameters were varied pairwise to create phase diagrams of parameter space. We previously submitted, and continue to include, a phase diagram comparing different values of myosin node activity and myosin node density (Figure 3C). The added phase diagrams investigate all other key parameters in the parameter table (Table 1). These analyses indicate that our two main findings are robust to changes in parameters: (1) Arp2/3 expansion is incapable of inducing a smooth and circular interface with a non-mechanosensitive actomyosin network, except with parameter values producing a network-wide increase to actomyosin contractility, and (2) Arp2/3 expansion induces a smooth and circular interface with a mechanosensitive actomyosin network, except under specific and explainable conditions (e.g. with values that excessively weaken Arp2/3 network growth and would thus elicit insufficient myosin activation).

Please see added Figure 4—figure supplements 1-6 and further explanation in the associated figure legends. Please see references to these figure supplements at lines 296 and 340.

3) Describe the model a bit more specifically at the beginning of the result section so the reader has a general sense of how it was implemented.

We have added more explanation at the beginning of the Results section, starting at line 126, and continue to refer to the Model Formulation section for full details.

4) Using physically relevant units in the text and the table would make it easier to follow and potentially more impactful. Or the authors should explain if there is a good reason to keep those non-physical units.

Our preference is to refer directly to properties of the simulations using units of the simulations. We consider our simulations to be their own experimental system. Our approach ensures that it is described as directly and accurately as possible. This approach also allows our simulation data to be more directly applied to a range of contexts. As acknowledged in the reviews, we do provide Table 1 which contains available equivalencies to in vitro and in vivo parameter values. Previously, we only referred to Table 1 in the Model Formulation section, but we now additionally refer to it at the beginning of the Results section (line 146).

5) Please provide Matlab files at resubmission.

A GitHub link is provided for the MATLAB files, and we continue to provide PDF files with annotated code as supplemental information. Please see line 452.

Reviewer #1:The authors attempt to understand the basis of smoothing interfaces between Arp2/3-based actin networks and actomyosin networks in the syncytial *Drosophila* embryo during pseudo-cleavage. This is an interesting question in the context of how actin networks are organized. To understand interface smoothing, the authors perform "node-based" simulations of the two actin networks. The simulation methods of the two networks follows previously published methods and are validated through comparisons with previous work. For the interface between these networks, the authors invoke some ad hoc rules that are poorly justified form a physical point of view. It is not clear, how much their results depend on this choice of interfacial dynamics.

We agree that a lack of understanding of the interface is a limitation of our study, and continue to state this limitation in the paper. However, our rules of node interaction across the interface are relatively simple and based on force balance between nodes directed toward each other because of net forces applied to them. Also, our results were robust to a range of parameter values affecting the interface. Please see added Figure 4—figure supplement 6 and further explanation in the associated figure legend.

Whereas the simulation results are quantitatively analyzed, there is essentially no quantification of the experiments, and the comparison between experiment and theory remains qualitative.

Unfortunately, smoothening was too challenging to consistently quantify in the experimental data due to variable shapes and intensities of myosin-positive structures within images, and variability of overall signal intensity between samples. To show that the control and Arp3 RNAi data were qualitatively reproducible across embryos, we now provide images of six additional embryos of each genotype. These examples also allow additional comparisons with the simulation results.

Please see Figure 3—figure supplement 1.

Eventually, the authors propose a set of ingredients that are sufficient to yield smooth interfaces in their simulations. However, a thorough understanding in terms of the physical mechanism underlying the simulation results remains elusive.

Please see response under “Essential Revisions”, above.

Reviewer #2:[…]Some of the parameters used in all the simulations are set to certain values but those values are often not discussed, and it is unclear to which extent the results of the simulations are sensitive to those parameters.

Thank you for this important recommendation. Please see response under “Essential Revisions”, above.

The code is provided only as a pdf file which prevents the reader to run the simulations themselves.The authors could test how sensitive their main results are to initial conditions, and to different parameter values by performing a sensitivity analysis (especially on the wild-type like cases)Please provide Matlab files at resubmission.

They have been provided. Please see response under “Essential Revisions”, above.

Things that could be clarified:The authors could explain a bit more how the initial rough patterns of Arp2/3 and acto-myosin zones are generated in the embryo. Is it signaling? Could it be self-organized from a random initial organization by the mechanics too?

Signaling is important for the initial patterning, but roles for mechanics are unknown. Specifically, Rho small G protein signaling pathways play essential roles in establishing the initial rough patterns of the cytoskeletal networks. Please see added text at lines 74 and 85-86.

They could describe the model a bit more specifically at the beginning of the result section so the reader has a general sense of how it was implemented.

Thank you for this suggestion. Please see revised text between lines 126-140.

Using physically relevant units in the text and the table would make it easier to follow and potentially more impactful. Or the authors should explain if there is a good reason to keep those non-physical units.

Please see response under “Essential Revisions”, above.